# Effects of High-Linear-Energy-Transfer Heavy Ion Radiation on Intestinal Stem Cells: Implications for Gut Health and Tumorigenesis

**DOI:** 10.3390/cancers16193392

**Published:** 2024-10-04

**Authors:** Santosh Kumar, Shubhankar Suman, Jerry Angdisen, Bo-Hyun Moon, Bhaskar V. S. Kallakury, Kamal Datta, Albert J. Fornace

**Affiliations:** 1Department of Oncology, Lombardi Comprehensive Cancer Center, Georgetown University Medical Center, Washington, DC 20057, USA; 2Department of Pathology, Georgetown University Medical Center, Washington, DC 20057, USA; 3Department of Biochemistry and Molecular & Cellular Biology, Georgetown University Medical Center, Washington, DC 20057, USA

**Keywords:** high-LET radiation, oxidative stress, DNA damage, senescence, senescence-associated secretory phenotype, intestinal tumor

## Abstract

**Simple Summary:**

Heavy ion radiation, found in outer space and used in some cancer treatments, can damage vital cells in the intestines. Studies using mice show that this type of radiation causes long-lasting stress, accelerated aging, and harmful changes in these cells, which can perturb gut function and increase the risk of developing cancer. Specialized mouse models were employed to monitor how these intestinal stem cells are affected over time after exposure to this radiation. We found that particle radiation caused more stress-induced damage and tumor incidence compared to photon radiation. Intestinal stem cells showed signs of aging and inflammation, which persisted for up to a year. This ongoing stress and damage also disrupt the gut barrier’s function and ability to absorb nutrients properly. The findings suggest that astronauts exposed to this radiation during deep space missions might face increased risks of gut dysfunction as well as increased cancer risk.

**Abstract:**

Heavy ion radiation, prevalent in outer space and relevant for radiotherapy, is densely ionizing and poses a risk to intestinal stem cells (ISCs), which are vital for maintaining intestinal homeostasis. Earlier studies have shown that heavy-ion radiation can cause chronic oxidative stress, persistent DNA damage, cellular senescence, and the development of a senescence-associated secretory phenotype (SASP) in mouse intestinal mucosa. However, the specific impact on different cell types, particularly Lgr5^+^ intestinal stem cells (ISCs), which are crucial for maintaining cellular homeostasis, GI function, and tumor initiation under genomic stress, remains understudied. Using an ISCs-relevant mouse model (*Lgr5*^+^ mice) and its GI tumor surrogate (*Lgr5*^+^*Apc*^1638N/+^ mice), we investigated ISCs-specific molecular alterations after high-LET radiation exposure. Tissue sections were assessed for senescence and SASP signaling at 2, 5 and 12 months post-exposure. Lgr5+ cells exhibited significantly greater oxidative stress following ^28^Si irradiation compared to γ-ray or controls. Both Lgr5^+^ cells and Paneth cells showed signs of senescence and developed a senescence-associated secretory phenotype (SASP) after ^28^Si exposure. Moreover, gene expression of pro-inflammatory and pro-growth SASP factors remained persistently elevated for up to a year post-^28^Si irradiation. Additionally, p38 MAPK and NF-κB signaling pathways, which are critical for stress responses and inflammation, were also upregulated after ^28^Si radiation. Transcripts involved in nutrient absorption and barrier function were also altered following irradiation. In *Lgr5*^+^*Apc*^1638N/+^ mice, tumor incidence was significantly higher in those exposed to ^28^Si radiation compared to the spontaneous tumorigenesis observed in control mice. Our results indicate that high-LET ^28^Si exposure induces persistent DNA damage, oxidative stress, senescence, and SASP in Lgr5^+^ ISCs, potentially predisposing astronauts to altered nutrient absorption, barrier function, and GI carcinogenesis during and after a long-duration outer space mission.

## 1. Introduction

The space radiation environment is typified by the presence of highly charged, high-energy (HZE) particles from the galactic cosmic ray (GCR) environment, as well as predominantly protons from solar particle events (SPEs) [1,2,3]. Exposure to ionizing radiation (IR) in space, as well as in environmental, occupational, and medical settings to some extent, typically involves the simultaneous action of various radiation types. Certain professions, such as airplane crews, can expose workers to a complex mix of γ-rays, neutrons, and protons [4]. Radiotherapy patients are treated typically with X-rays, but other particle beams, such as protons, neutrons, and carbon ions have also been used [5,6,7,8,9,10,11]. Long-duration space trips such as a Mars mission are expected to expose astronauts to densely ionizing high-linear-energy-transfer (LET) radiation in the dose range totaling 0.3–0.5 Gy or more [2,12]. These high-LET radiations consist of various charged particles like ^56^Fe, ^28^Si, and ^12^C, all with a broad range of energies. These high-LET radiations can cause considerable normal tissue damage, resulting in acute or chronic toxicities [13,14,15,16]. The best overall estimate of the probability of direct ionization at a given site in a molecule, e.g., in DNA—such as the sugar, phosphate, or DNA base—is provided by the number of valence electrons at that site [17]. The complexity and variability of radiation exposure in different contexts underscore the need for continued research and monitoring to understand better the potential health risks associated with HZE-ion irradiation [18,19].

Intestinal tissue is highly proliferative and particularly sensitive to IR showing oxidative stress, DNA damage, and other pathologies, including colorectal cancer [15]. Lgr5 expressing ISCs normally reside at the crypt base where they divide and differentiate into various cell types with migration along the crypt-villi axis [20]. These various differentiated cells contribute in maintaining the intestinal homeostasis and overall health on an individual [16,20]. Notably, Paneth cells, derived from Lgr5^+^ ISCs like other epithelial cell lineages in the small intestine, secrete antimicrobial peptides to defend the host against enteric pathogens, also serves as sensors of nutritional status, and its dysfunction has been linked to the development of GI tract inflammation [21,22,23]. Healthy Lgr5^+^ stem cells are essential for the proper function of Paneth cells because the interactions between Lgr5^+^ cells and Paneth cells create a supportive microenvironment that promotes stem cell activity, allowing for the continuous regeneration of the intestinal epithelium. Any disruption in the health or function of Lgr5^+^ cells can also impair Paneth cell function, potentially leading to compromised gut integrity and an increased susceptibility to gastrointestinal disorders.

HZE particles are known to cause clustered damage and release of short DNA fragments, which are difficult to repair accurately [17,24,25,26,27]. Consequently, relative to low-LET IR, high-LET heavy ions can cause greater damage to Lgr5^+^ ISCs, leading to decreased numbers, reduced self-renewal, and compromised differentiation potential [14,16]. The p38/MAPK signaling pathway regulates many cellular responses to adverse stimuli, such as DNA damage, oxidative stress, and inflammatory cytokine secretion. Moreover, it is an essential regulator of the NFκB pathway, responsible for the expression of genes involved in inflammation, cell survival, and apoptosis [28,29,30,31]. The adverse effects of radiation on the GI tract have been reported by our group as well as by other investigators documenting histopathological and functional alterations in the GI tract post-irradiation [13,15]. In our previous study, we reported that the high-LET heavy-ion irradiation initiates a damaging cascade involving oxidative stress, DNA damage, senescence, and acquisition of SASP [16]. Senescence/SASP cells can trigger a senescence-inflammatory response (SIR) pathway that can further exacerbate oxidative stress and DNA damage through the accumulation of pro-inflammatory factors creating a vicious cycle, perpetuating ongoing cellular dysfunction and contributing to the progressive deterioration of intestinal tissue function [14,15,16]. This study aims to elucidate the differential impact of photon (γ-ray) radiation and high-LET ^28^Si, on ISCs, with a focus on Lgr5^+^ ISCs, which are critical for maintaining gastrointestinal (GI) homeostasis and function. By utilizing an ISCs-relevant mouse model (*Lgr5*^+^ mice) and a GI tumor surrogate model (*Lgr5*^+^*Apc*^1638N/+^ mice), we investigated the long-term effects of radiation exposure on oxidative stress, DNA damage, cellular senescence, and the development of a senescence-associated secretory phenotype (SASP) that may predispose astronauts to GI dysfunction and carcinogenesis during and after prolonged space missions.

## 2. Materials and Methods

### 2.1. Animals and Irradiations

Male Lgr5-EGFP-IRES-creERT mice (*Lgr5*^+^ mice) were purchased (Jackson Laboratory Stock No: 008875, Bar Harbor, ME, USA) and bred with C57BL6 female mice and male pups were genotyped as described previously [20] and randomly assigned to the experimental groups. The *Apc*^1638N/+^ mouse colony was maintained at GU as described earlier [32]. For the tumorigenesis study, female *Lgr5-EGFP-IRES-CreERT2* mice were crossed with male *Apc*^1638N/+^ and genotyped as described earlier [20,32] to generate double heterozygote *Lgr5*^+^*Apc*^1638N/+^ mice. Male *Lgr5*^+^*Apc*^1638N/+^ mice (n = 15/group, 6–8 weeks old) were exposed to whole-body ^28^Si ion radiation (energy: 300 MeV/nucleon; LET: 69 keV/μm; dose: 50 cGy and dose rate ~50 cGy/min) using the simulated space radiation facility at NASA Space Radiation Laboratory (NSRL) in Brookhaven National Laboratory (BNL), and control mice were sham-irradiated. The selection of a 50 cGy dose was based on our previous tumorigenesis studies [33,34,35]. Accurate dose delivery was ensured through beam calibration and dosimetry by NSRL physicists, with variability between exposures maintained within ±2.5%. For further details on beam calibration and dosimetry please refer to https://www.bnl.gov/nsrl/userguide/dosimetry-calibration.php (15 August 2024). All mice were shipped a week before radiation exposure for acclimatization at the BNL animal facility and returned a day after radiation exposure to the GU animal facility in a temperature-controlled environment to minimize transportation-related stress. Mice were housed in an air- and temperature-controlled room with a 12 h dark and light cycle maintained at 22 °C in 50% humidity at the GU as well as at the BNL animal care facility. All the mice were provided food and filtered water *ad libitum*. Studies on *Lgr5*^+^ mice were conducted up to one year post-exposure, whereas tumorigenesis study in *Lgr5*^+^*Apc*^1638N/+^ mice was conducted at 5 months after irradiation as described earlier [35]. All animal procedures were approved by the Institutional Animal Care and Use Committees at BNL (Protocol#345, 12 October 2021) and GU (Protocol#2016-1129, 1 August 2021). Our research followed the Guide for the Care and Use of Laboratory Animals, prepared by the Institute of Laboratory Animal Resources, the National Research Council, and the U.S. National Academy of Sciences. 

### 2.2. Isolation of Lgr5^+^ ISCs

Lgr5^+^ cells were isolated as described earlier with minor modifications [16]. Mice were euthanized at 60 d or 1 year after radiation exposure, small intestines were incised, flushed, inverted to the lumen facing outward, chopped into 10 mm pieces and washed thoroughly with ice-cold phosphate-buffered saline (PBS). Tissues were incubated with 2 mM EDTA in PBS for 20 min, then washed vigorously twice with PBS. The supernatant containing villus cells was discarded, and the tissue cell dissociation enzyme [STEMxyme^®^ 2 Collagenase/Neutral Protease (Dispase) Cat# LS004112, Worthington Biochemical Corporation, Lakewood, NJ, USA] 1 mg/mL and 0.1 mg/mL DNase 1 (Cat# 89836, Thermo Scientific, Waltham, MA, USA) in HBSS (Cat# 14175103, Thermo Scientific) were added and incubated for 20 min at 37 °C. The supernatant was discarded and cells were released with vigorous shaking in cold HBSS. The cell suspension was passed through a 70 μm strainer, centrifuged, washed twice with cold HBSS and resuspended in HBSS containing 2% FBS for ISCs sorting. Cells were sorted on FACS ARIA IIU (Becton Dickinson, San Jose, CA, USA) sorter using FACSDIVA software version 9.0. Proper electronic gates of side scatter and forward scatter parameters were set to exclude debris and doublet cells. Viable cells were gated for the negative staining for SYTOX™ Blue (cat#S34857, Life Technologies, Frederick, MD, USA) to exclude dead cells with 405 nm laser through 450/40 band pass filter. GFP signal was collected with 488 nm laser through a 530/40-band pass filter.

### 2.3. Intracellular and Mitochondrial Reactive Oxygen Species (ROS) Detection

Intracellular and mitochondrial ROS were determined using CellROX™ Deep Red Flow Cytometry Assay Kit (cat# C10491, Life Technologies) and MitoSOX red (cat# M36008 Life Technologies) according to the manufacturer’s instruction, respectively. Briefly, cells were incubated with 500 nM CellROX™ Deep Red or 2 µM mitoSox red for 30 min at 37 °C, then washed with PBS and fluorescence intensity was assessed by a BD LSRFortessa flow cytometer (Becton Dickinson, San Jose, CA, USA). Cells were counter-stained with SYTOX™ Blue (cat#S34857, Life Technologies) to exclude dead cells while acquiring data. Thresholds were adjusted using unstained cells for CellRox or MitoSox before obtaining the fluorescence intensity values using FCS express software version 7 (De Novo Software, Pasadena, CA, USA). Average fluorescent intensity was calculated from three independent experiments using three different mice per experiment. CellROX stained cells were paraformaldehyde fixed, cytospun onto a glass slide, and visualized under a fluorescent microscope to further validate the observation.

### 2.4. β-Galactosidase Staining

β-galactosidase staining of intestinal sections was performed as described elsewhere [20]. Briefly, the intestine was flushed with ice-cold PBS to remove feces. Freshly harvested one-inch sections of jejunum were isolated and immediately incubated for 2 h in ice-cold fixative (1% formaldehyde, 0.2% glutaraldehyde and 0.02% NP40 in PBS) on a rocking platform at 4 °C. The fixative was removed and the tissues were washed twice in PBS for 20 min at room temperature on a rolling platform. The tissues were incubated with β-galactosidase substrate (5 mM potassium ferricyanide, 5 mM potassium ferrocyanide, 2 mM MgCl_2_, 0.02% NP40, 0.1% sodium deoxycholate and 1 mg/mL X-gal (Cat#15520018, Life Technologies) in PBS (pH 5.6) in the dark overnight at room temperature. The tissues were washed twice in PBS for 20 min at room temperature on a rolling platform and then fixed overnight in formalin at 4 °C in the dark on a rocking platform. The stained tissues were transferred to tissue cassettes and paraffin blocks prepared using standard methods. Tissue sections (~5 μM thickness) were prepared and counterstained with nuclear fast red (cat# H-3403, Vector Laboratories, Inc., Newark, CA, USA).

### 2.5. In Situ Immunostaining

Paraffin-embedded tissue sections were deparaffinized and sequentially rehydrated and antigens were retrieved using boiling pH 6.0 citrate buffer (Electron Microscopy Sciences, Hatfield, PA, USA). Samples were treated with primary antibodies (anti-Lgr5-GFP, anti-lysozyme, anti-IL1β, anti-IL1R, anti-phospho-p38, and anti-phospho-p65) overnight at 4 °C and signals were detected using DAB substrate provided in the Mouse- and Rabbit-Specific HRP/DAB IHC detection kit (Cat#AB236466; Abcam, Cambridge, MA, USA) according to the manufacturer’s instruction. Sections were counterstained with hematoxylin, sequentially dehydrated and mounted using a Permount mounting medium (Cat# SP15-100, Fisher Chemical, Waltham, MA, USA). Slides were visualized under a bright field microscope (Olympus) and images were captured. For immunofluorescence, samples were incubated with antibodies (anti-γH2AX+anti-p16+anti-Lgr5-GFP or anti-IL8+anti-p21+anti-Lgr5-GFP) overnight at 4 °C, washed and incubated with secondary antibody IgG-conjugated with Alexa Fluor 488 (green), 594 (Red) and 647 (Far red) for 1 h at room temperature. Nuclei were counter-stained with DAPI. Slides were visualized and images were captured using an Olympus fluorescent microscope.

### 2.6. mRNA Expression Analysis Using Real-Time PCR Assays

Total RNA was extracted from flow-sorted flash-frozen ISCs using RNeasy mini kit (Cat# 74104, Qiagen, Germantown, MD, USA) and was reverse transcribed to cDNA using RT2 First Strand Kit (Cat# 330421; Qiagen, Valencia, CA, USA). Oxidative stress PCR array was performed using RT^2^ Profiler™ PCR Array Mouse Oxidative Stress and Antioxidant Defense (Cat#PAMM-065ZD-12; Qiagen) as per the manufacturer’s instruction. Real-time PCR for genes involved in SASP or senescence inflammatory response (SIR) signaling was performed using specific primers as listed in Appendix A (Eurofins Genomics, Louisville, KY, USA) and SYBR green master mix (Qiagen) on CFX96 real-time instrument (Bio-Rad, Hercules, CA, USA) using the temperature settings: 95 °C for 5 min and then 40 cycles of 95 °C for 15 s and 58 °C for 1 min. We employed the ΔΔCt method, in which ΔCt was calculated using the *Gapdh* reference gene and ΔΔCt was calculated relative to the control group as described earlier [15]. Changes in gene expression data are presented as fold change relative to control ± standard error of the mean (SEM).

### 2.7. Imaging, Quantification, and Statistical Analysis

To ensure the specificity of the immunostaining, appropriate controls were conducted alongside the experimental samples. In each section, randomly selected fields of vision for normal mucosa or tumors were captured using cellSens Entry v1.15 (Olympus Corp, Center Valley, PA, USA) for both immunohistochemistry and immunofluorescence as described earlier [36]. The average DAB pixel density or fluorescent intensity was measured from 10–15 images for each marker. Statistical significance between the two groups was determined using a two-tailed paired Student’s t-test, and data were graphically presented as mean ± standard error of the mean (SEM). All data were statistically analyzed, with a *p*-value of 0.05 or less considered statistically significant. 

### 2.8. List of Antibodies

IL6 (cat#ab7737, dilution 1:200); IL1R (cat#ab124962, dilution 1:200) Abcam, Cambridge, MA, USA; γH2AX (cat# 4418-APC-100, dilution 1:50), Trevigen Gaithersburg, MD, USA; p16 (cat#SC-1661, dilution 1:200), p-p38(cat#SC-166182, dilution 1:200), p-p65 (cat# SC33039, dilution 1:100), β-catenin (cat# SC-7963, dilution 1:200) Santa Cruz Biotechnology, Dallas, TX, USA; IL8 (cat#orb229133, dilution 1:200) Biorbyt LLC, San Francisco, CA, USA; p21 (cat#05-345, dilution 1:100), phospho-Histone H3(Ser10) (Cat#09-797, dilution 1:50)Millipore, Burlington, MA, USA; GFP(cat#600-101-215M, dilution 1:200) Rockland Immunochemicals, Inc., Limerick, PA, USA; Lyzozyme (cat# A009902-2, dilution 1:300) Dako Santa Clara, CA, USA; IL1b (cat#AF-401-NA, dilution 1:200) Minneapolis, MN, USA.

## 3. Results

### 3.1. Assessment of ROS in Lgr5 Expressing ISCs after ^28^Si Irradiation in Mice

Intracellular ROS was assessed using fluorescent probe CellROX and flow cytometer in sorted ISCs. Our data show increased ROS in ISCs indicated by a right shift in the histogram two months after irradiation (Figure 1A). Quantification of flow cytometry data showed a radiation quality-dependent increase in ROS (*p* = 0.021 γ-ray; *p* = 0.04 ^28^Si) compared to the control (Figure 1B). Findings from flow cytometry were also validated using fluorescence imaging where higher CellROX fluorescence was observed in ^28^Si than in γ-rays or control (Appendix A). To assess the mitochondrial superoxide in flow-sorted ISCs, we used MitoSOX Red and analyzed it in a flow cytometer. Increased mitochondrial superoxide was observed in radiation-exposed ISCs (Figure 1C). Flow cytometry data show significantly higher mitochondrial superoxide in the ^28^Si-irradiated group (*p* = 0.006, ^28^Si) than the control group (Figure 1D). However, an insignificant increase in Mitosox level was observed in the γ-ray group compared to the control. As expected, higher cellROX or Mitosox staining was observed in ^28^Si relative to γ-ray exposed ISCs (Figure 1B,D). RT^2^ Profiler™ PCR Array data showed differential regulation of oxidative stress genes in radiation-exposed ISCs. The data show significantly higher expression of oxidative stress-responsive genes (*Sod1*, *Prdx6*, *Ercc6*, *Gpx5* and *Noxo1*) in the ^28^Si-irradiated group compared to the control (Appendix A).

### 3.2. ^28^Si Ion Radiation-Induced DNA Damage and Cell Proliferation

Apart from IR, increased ROS is also known to induce DNA double-strand break (DSBs) in IEC. We performed γH2AX and Lgr5-GFP co-staining to assess DNA DSBs in intestinal tissue. Amplified γH2AX foci in ^28^Si-irradiated samples indicate the continued presence of DNA DSBs in the ISCs and non-stem cells after heavy ion radiation (Figure 1E,F). Our data showed a modest decrease in Lgr5 staining in intestinal cells when exposed to ^28^Si compared to the control (Figure 1G). Oxidative stress regulates survival and cell proliferation depending on the threshold level. We stained the tissue section for phospho-Histone H3, a known cell proliferation marker, and showed significantly increased phospho-Histone H3 staining in irradiated (*p* < 0.001 for ^28^Si, *p* = 0.014 for γ-ray) samples (Figure 2A). Consistent with our earlier findings, these data indicated the persistence of sub-lethal DNA damage and enhanced cell proliferation in the ^28^Si-irradiated group compared to the γ-ray or control group.

### 3.3. Lgr5^+^ ISCs and Paneth Cell Senescence after Heavy-Ion ^28^Si Radiation

IR-induced increases in ROS and persistent DNA damage are known to induce premature cellular senescence. While senescence is common among cells that have the capacity to divide, like Lgr5^+^ ISCs, differentiated cells such as Paneth cells can also accumulate DNA damage over time after exposure to genotoxic stressors and undergo senescence, potentially losing their capacity to de-differentiate leading to impaired recovery [37,38]. To determine whether ^28^Si induces senescence in IEC, irradiated intestinal tissue was stained for senescence associated-β-galactosidase (SA-β-gal) activity. The results reveal a substantial increase in SA-β-gal positive cells in both ^28^Si (*p* < 0.001) and γ-ray (*p* < 0.001) irradiated intestinal sections, whereas control mice show relatively low SA-β-gal activity (Figure 2B,C). Moreover, intestinal sections were co-stained with SA-β-gal and lysozyme (Paneth cell marker) to identify the specific cell types impacted after IR. The data show a higher level of Paneth cell senescence in intestinal sections in irradiated groups compared to control. The increase in Paneth cell senescence was significant in ^28^Si (*p* < 0.001) or γ-ray (*p* < 0.001) compared to control (Figure 2D,E). The data demonstrate an almost two-fold increase in Paneth cell senescence for ^28^Si relative to the γ-ray-irradiated group (Figure 2E). We further co-stained SA-β-gal and Lgr5 to detect senescent ISCs. A higher number of Lgr5 positive senescent ISCs was observed in ^28^Si compared to γ-ray or control. The *Lgr5* positive stem cell senescence was significantly higher in the ^28^Si-irradiated (*p* < 0.001) group compared to the control (Figure 2F,G). Consequently, these data indicated increased senescence in both ISCs and differentiated Paneth cells that could potentially impact GI function.

### 3.4. Increased SASP Markers and Pro-Inflammatory MAPK/NFκB Signaling after ^28^Si Radiation

We further examined the expression of senescence-associated inflammatory genes that changed after exposure to high-LET radiation. Sorted ISCs from 2 months after radiation exposure were analyzed with quantitative reverse transcriptase-PCR (qRT-PCR) analysis. Expression of a small subset of genes was higher in mice irradiated with ^28^Si relative to unirradiated control. However, some of these genes were not expressed at higher levels in γ-ray-irradiated groups. The data show SIR genes (*Atf5*, *CD40*, *Cxcl1*, *Cxcl5*, *Faim2*, *Lpo*, *Opg*, *Pla2g2a*, *Ptges*, *Sox17*, *Tlr1*, and *Troy*), proliferation genes (*Cyclin D1* and *Cyclin D2*) and SASP genes (*Cxcl2*, *Il1β*, and *Il6*), along with the senescence-associated gene (*p19Arf*, *p16* and *p21*), were markedly increased 2 months after ^28^Si irradiation in ISCs (Appendix A). To demonstrate that senescent cells acquired SASP after irradiation, intestinal tissue sections were co-stained with Lgr5-GFP, IL8, and p21. The data show an overall increase in IL8 or p21 expression with increasing LET compared to control at 2 months post-irradiation (Figure 3A). The data also demonstrate a higher number of p21 positive cells with IL8 and Lgr5-GFP expression with increasing LET indicating a higher number of senescent ISCs acquired SASP two months after irradiation. Lgr5-GFP expression analysis showed trivial decreased expression with increasing LET at two months post-irradiation (Figure 3A,B). To further demonstrate SASP, we performed immunostaining for IL1β and IL1R in β-gal-stained slides. The data show higher levels of IL1β and IL1R in ^28^Si compared to γ-ray or control irradiated samples (Figure 3C–F). Immunostaining of intestinal sections shows significantly higher β-catenin expression level while lower Lgr5-GFP expression in the ^28^Si-irradiated group relative to the γ-ray or control group (Appendix A). To examine whether radiation-induced DNA damage response, senescence and SASP have an impact on a stress MAPK signaling pathway, immunohistochemistry assays were performed for phosphorylated forms of p38 and NFκB. The data demonstrated higher phospho-p38 expression in irradiated groups compared to the control (Figure 3G,H). Phospho-p38 is known to activate its downstream target NFκB, while NFκB is known as a regulator of IL6 and IL1β in response to IR [29,30,39,40,41]. The data show elevated levels of activated NFκB expression in irradiated samples compared to the control. Notably, the data indicated a higher level of p-NFκB expression in ^28^Si compared to the γ-ray (Figure 3I,J).

### 3.5. Persistent DNA Damage, Senescence and SASP Marker in Intestine Long-Term after Heavy-Ion Radiation

Our earlier study using isolated ISCs from heavy-ion-irradiated mice demonstrated higher oxidative stress, DNA damage, senescence and SASP acquisition at 2 months post-exposure [16]. Here, we demonstrate the persistence of chronic stress marked by higher oxidative stress, DNA damage, senescence and acquisition of SASP chronic stress in ISCs of heavy-ion-irradiated mice up to 12 months post-irradiation. We assessed DNA double-strand break and senescence in the same section using γH2AX, p16, and Lgr5-GFP anti-antibodies in the co-staining experiment. We observed an increase in the average number of γH2AX foci in the irradiated groups compared to the control at 5 and 12 months post-irradiation (Figure 4A). An average number of γH2AX foci in ^28^Si were observed higher compared to γ-ray at 5 and 12 months post-irradiation. The results also indicated higher p16 staining after irradiation relative to the control group. The data show higher p16 staining in ^28^Si compared to γ-ray or the control group. Lgr5-GFP co-staining enables us to identify senescent ISCs containing DSB. The data showed a higher number of p16 positive nuclei with Lgr5-GFP staining and γH2AX foci in ^28^Si compared to γ-ray or control group suggesting a higher number of senescent ISCs with DSB in irradiated samples relative to the control (Figure 4A–D). We compiled the DSB data for 2-, 5- and 12-month samples and represented them graphically. The data demonstrated a significant increase in DSB with increasing LET as well as increasing time post-irradiation (Figure 4E). We further examined IL8, p21 and Lgr5-GFP expression at 5 m and 12 m post-irradiation, and found a higher level of IL8 as well as p21 expression in ^28^Si relative to the γ-ray or control irradiated samples (Figure 5A–D).

### 3.6. IR-Induced Accelerated Aging Is Associated with Altered Nutrient Absorption and Barrier Function

Accelerated aging is associated with several pathological alterations in intestinal homeostasis. Barrier function and nutrient absorption are prominent functions of the intestine to maintain the healthy lifestyle of an individual. Here, we have investigated the expression of transcripts involved in nutrient absorption through qPCR analysis. The results demonstrated altered expression of glucose transporter (*Slc2a2*, *Slc2a5*, and *Slc5a1*), gut hormone and Na/H exchange (*Cck*, *Gip*, and *Slc9a3*) and cholesterol and fatty acid transporters (*Npc1*, *Npc1/1*, and *Slc27a4*) in the irradiated group relative to control after 2 months (Figure 6A–C) or 12 months (Figure 6D–F). For the barrier function study, we detected the level of citrulline and I-FABP in the serum. The data show a significant decrease in citrulline (the marker for intestinal integrity) levels in the serum of ^28^Si samples relative to the control up to 12 months post-exposure (Figure 6G,I). We showed a significant increase in I-FABP (the marker for intestinal injury) level in ^28^Si relative to control up to 12 months after irradiation. However, insignificant changes in the citrulline or I-FABP levels were observed in γ-ray relative to the control. Overall, these results suggest that high-LET radiation alters nutrient absorption and barrier function after 2 and 12 months post-irradiation (Figure 6G–J).

### 3.7. Increased Senescence and SASP Factors in Intestinal Tumor in Lgr5^+^Apc^1638N/+^ Mice after ^28^Si Radiation

We also have previously demonstrated increased SASP and tumorigenesis in the *Apc*^1638N/+^ mice model after heavy-ion radiation [35]. Here, we wanted to detect the effect of ^28^Si ions in ISCs in normal and tumor tissue from the *Lgr5^+^Apc*^1638N/+^ mice model. Irradiated male *Lgr5*^+^*Apc*^1638N/+^ mice showed the highest tumor incidence in ^28^Si (7.83 ± 1.10, *n* = 9) relative to γ-ray (4.73 ± 0.27, *n* = 15) or control (2.88 ± 0.17, *n* = 15) group (Appendix A). We found enhanced DNA damage and SASP in ISCs from ^28^Si relative to the γ-ray or control group, which confirms our earlier reports (Figure 7A–D). Both normal and tumor tissue were examined for γ-H2AX, p16, and Lgr5-GFP expression through immunostaining. The results show a higher number of γH2AX foci and enhanced p16 expression in the ^28^Si-irradiated group relative to the γ-ray or control group both in tumor or normal tissue (Figure 7B and Appendix A). The level of p16 expression and γH2AX foci was significantly higher in tumors relative to normal tissue across the treatment groups (Figure 7B and Appendix A). To further examine SASP acquisition, tissue sections were co-stained with IL8, p21, and Lgr5-GFP and analyzed under fluorescent microscopy. The data demonstrate increased IL8 and p21 expression after ^28^Si compared to the γ-ray or control group in normal as well as tumor tissue. These pieces of evidence suggest the greater acquisition of SASP in ^28^Si compared to the γ-ray or control group in the tumor as well as normal tissue (Figure 7C,D and Appendix A), whereas a lower level of Lgr5-GFP expression was observed after irradiation in normal as well as in tumor tissue (Figure 7D, Appendix A).

## 4. Discussion

Our findings demonstrate the potential for heavy-ion radiation exposure, such as ^28^Si ion, to induce oxidative stress in ISCs at higher levels relative to γ-ray exposure. This is consistent with our earlier finding where we have shown increased ROS after 2 months of ^56^Fe irradiation in ISCs [16]. We also found a corresponding decrease in the expression of antioxidant enzymes in the radiation-exposed ISCs, suggesting a disruption in the balance between pro-oxidant and antioxidant defense systems. The increased γH2AX foci in samples irradiated with ^28^Si ions suggest the presence of substantial DNA damage even with heightened cell proliferation. These findings are consistent with our earlier report with γ-ray or ^56^Fe particle radiation in intestinal epithelial cells [15]. Given the stochastic nature of physicochemical and chemical processes in radiation-DNA interactions, previous findings suggest that the quantity and spatial distribution of ionization events may underestimate both the extent and complexity of DNA lesions induced by high-LET particles [42]. Research, including Monte Carlo simulations and experiments, reveals that DNA damage, such as DSBs and non-DSB lesions, tends to cluster within localized regions of the chromosome, often within 10–20 megabase pairs. This clustering can overwhelm DNA repair mechanisms, leading to more severe mutations or chromosomal issues, affecting DNA damage response pathways [17,24,25,26,42]. Furthermore, our data showed a modest decrease in Lgr5-GFP expression in intestinal cells when exposed to ^28^Si, suggesting that the ISCs can be affected by the radiation quality. Similar observations in Lgr5 expression after IR exposure were seen as reported earlier [36].

Radiation is known to induce oxidative stress and premature cellular senescence in intestinal epithelial cells [14,15,16]. Lgr5^+^ ISCs are essential for the continuous renewal and homeostasis of intestinal cells, whereas differentiated cells like Paneth cells are vital for maintaining the functional integrity of the GI tract. Moreover, Paneth cells have a longer life span (~60 days), and they provide essential signals and factors necessary for Lgr5^+^ ISCs maintenance and proliferation. The analysis of specific cell types showed that both Paneth cells and ISCs had higher levels of senescence in the ^28^Si exposed group compared to the exposed γ-ray or control group, indicating that ^28^Si particles are more effective at inducing premature senescence than γ-rays. Several studies have shown the acquisition of SASP after HZE-ion irradiation [14,15,16]. Higher expression of IL8 and IL1β in senescent cells indicates the acquisition of SASP as reported earlier [15,16,43]. Here, our data showed higher SASP acquisition in ^28^Si relative to the γ-ray or control group. Interestingly, data on the long-term study up to one year after irradiation show the persistence of senescence and SASP in ISCs. Arguably, cells may enter a vicious cycle of oxidative stress, DNA damage, senescence, and SASP in an adverse positive feedback loop [14,16,44,45,46]. Overall, our findings on Paneth cell senescence/SASP also suggest that altered secretory profile in these cells can create a chronic inflammatory environment in the gut, disrupting the local microenvironment and promoting further cellular damage and senescence in neighboring cells, including Lgr5^+^ ISCs.

SASP mediators like IL-1β mediate activation of non-muscle myosin light chain kinase and mediate epithelial barrier dysfunction in gut epithelial cells [47,48]. Evidence suggests multiple signaling pathways drive the ROS, DNA damage and SASP after radiation injury [29,40,41,49]. The MAPK/NFκB signaling pathway is an important cellular mechanism involved in the response to radiation-induced damage [39,40,41,50]. It has been shown that the MAPK/NFκB pathway is involved in controlling these processes in response to radiation [28,29,39,40,50,51]. Data on phosphorylated forms of p38 or NFκB, demonstrating enhanced expression of these molecules in response to ^28^Si compared relative to controls, indicate higher levels of activated p38 and NFκB. Our data suggest that the MAPK/NFκB signaling axis may play an important role in regulating stress responses and inflammatory cytokine secretion. Higher IL1β and β-catenin expression is linked to increased abnormal barrier function in intestinal epithelial cells [47].

Radiation-induced accelerated aging has been shown to affect intestinal barrier function and nutrient absorption [15,31,45,52]. The results of this study suggest that IR, particularly ^28^Si ions, can induce persistent alterations in the expression and activity of molecules involved in nutrient absorption, gut hormone production, and Na/H exchange as well as cholesterol and fatty acid transporters. Altered citrulline (intestinal integrity marker) and IFABP (intestinal injury marker) levels in serum after ^28^Si relative to control up to 12 months post-irradiation indicates long-term damage to the gut mucosa. Intestinal barrier dysfunction can lead to altered gut homeostasis and an elevated risk of early-onset colorectal cancer [53]. Our data further demonstrate the higher GI tumorigenic potential of high-LET compared to low-LET irradiation. While investigating the effects of ^28^Si on ISCs in normal intestine and tumor samples, results showed a significant increase in senescence and SASP in both. This was exemplified by increased expression of γH2AX foci, p16 or p21, and IL8 markers [54]. Additionally, Lgr5-GFP expression was found to be decreased after irradiation relative to the control group in both normal and tumor tissue [29]. These results suggest that space radiation can significantly affect stem cell biology, leading to accelerated senescence and risk of tumorigenesis [55]. Overall, this study provides novel insights into the specific vulnerabilities of ISCs and Paneth cells to high-LET radiation, highlighting the potential long-term health implications for astronauts. The persistence of oxidative stress, DNA damage, senescence, and the development of SASP after ^28^Si radiation suggests that these cells are trapped in a vicious cycle after ^28^Si irradiation (Figure 7E). Further, the sustained activation of the p38 MAPK/NF-κB pathway following ^28^Si irradiation underscores the importance of developing targeted protective strategies to mitigate radiation-induced damage in the GI tract, ultimately contributing to the safety and success of future space exploration.

## 5. Conclusions

Our findings provide experimental evidence unraveling the molecular underpinnings of how high-LET heavy-ion radiation in contrast to low-LET γ-ray can induce persistent oxidative stress, DNA damage and senescence even long-term after exposure. Modeling of persistent DNA damage foci using Monte Carlo calculations can provide further insight into the complexity and persistence of DNA damage induced by high-LET radiation. By understanding the mechanisms behind radiation-induced senescence and the senescence-associated secretory phenotype (SASP), we can gain a better understanding of how HZE exposure is associated with long-term adverse health consequences, including increased risks of cancer, inflammation, and tissue dysfunction. Overall, our study highlights the long-term risk associated with gut health and GI carcinogenesis after space radiation exposure and warrants further research into the mechanisms of radiation-induced senescence in order to develop effective countermeasures.

## Figures and Tables

**Figure 1 cancers-16-03392-f001:**
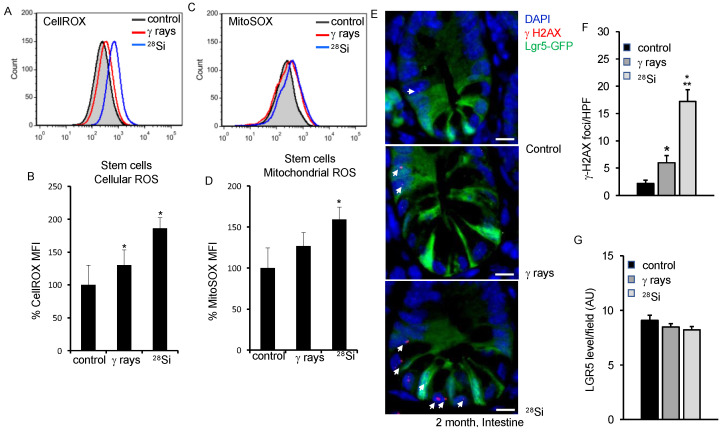
Heavy ion ^28^Si irradiation leads to persistent ROS and DNA damage in Lgr5^+^ ISCs two months after exposure. (**A**) Representative flow cytometry histogram showing increased CellROX fluorescence intensity in ISCs after ^28^Si radiation. (**B**) Quantifying fluorescence intensity data from five mice are presented as percent change in mean fluorescence in irradiated samples relative to controls. (**C**) Representative flow cytometry histogram showing increased MitoSOX fluorescence intensity in ISCs after ^28^Si radiation. (**D**) Quantified fluorescence intensity data from five mice is presented as a percent change in mean fluorescence in irradiated samples relative to control. (**E**) Representative IF images of intestinal sections co-stained for γH2AX and Lgr5-GFP showing DNA damage as foci. Nuclei were counterstained with DAPI. Scale bar 10 μm. (**F**) Images were quantified from 10 FOVs and analyzed statistically. Bar graph representing increased average γH2AX foci/HPF in the irradiated group. (**G**) Graphical representation of Lgr5-GFP mean fluorescent intensity showing a modest decrease expression after radiation relative to control. *, significant relative to control; **, significant relative to γ-rays.

**Figure 2 cancers-16-03392-f002:**
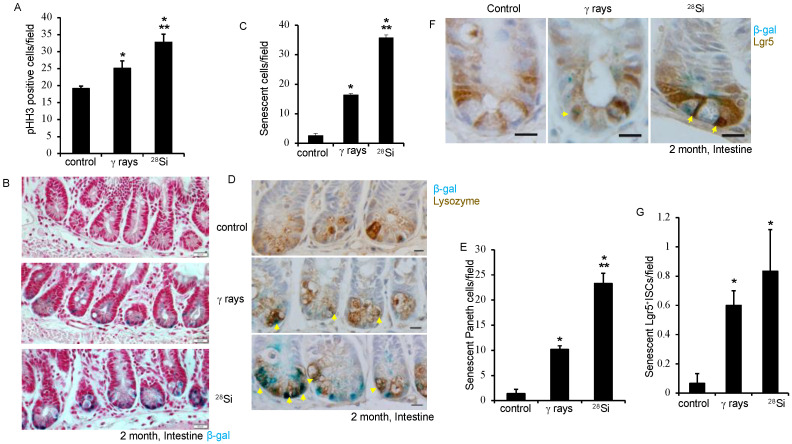
^28^Si ion exposure promotes crypt cell senescence 2 months after irradiation. Intestinal tissue sections were stained with phospho-Histone H3 (PHH3) to asses proliferation after irradiation. (**A**) Bar graph showing average PHH3 positive cells in each treatment group. (**B**) Representative images from each group showed increased SA-β-Gal positive cells (Blue color) at the crypt base after irradiation. Nuclei were counterstained with nuclear red stain. (Scale bar, 20 μm.) (**C**) SA-β-Gal positive cells were counted from at least 10 FOVs from each treatment group, analyzed statistically and represented in the form of a bar graph. (**D**) Senescent cells were detected using SA-β-Gal staining, and sections were co-stained with an anti-lysozyme (Paneth cell markers) antibody. Representative images showing Paneth cells (brown) senescence (blue) 2 months after irradiation. (Scale bar 20 μm.) (**E**) Average SA-β-Gal positive Paneth cells presented as a bar graph. (**F**) Representative images showed increased ISCs senescence after ^28^Si exposure. (Scale bar, 10 μm.) (**G**) Average SA-β-Gal positive ISCs presented in the form of a bar graph. Average count of senescent cells determined from at least 10 different FOVs, analyzed statically represented in the form of a bar graph. Nuclei were counterstained with hematoxylin. *, significant relative to control; **, significant relative to γ-ray.

**Figure 3 cancers-16-03392-f003:**
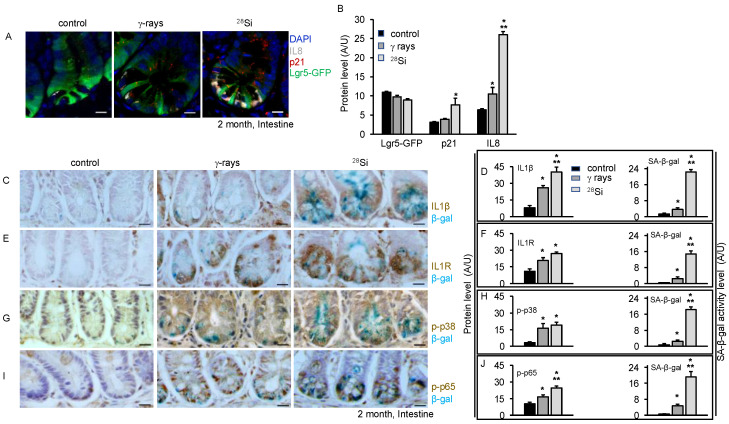
Heavy ion ^28^Si promotes accelerated SASP in mouse intestinal crypt 2 months post-exposure. (**A**) Fluorescent images showing IL8 (grey), p21 (red), Lgr5-GFP (green) in intestine sections. Nuclei were counterstained with DAPI. (**B**) Bar graph showing average pixel intensity/field of Lgr5-GFP, IL8 and p21. (**C**) Representative images show IL1β (brown) and senescent cells (blue) in intestinal tissue sections. (**D**) Graphical representation average DAB or β-Gal stain pixel intensity was quantified from at least ten different FOVs. (**E**) Representative images show senescent cells (blue) with IL1R (brown) expression in intestine sections. (**F**) Bar graph showing average pixel intensity of IL1β (left panel) or β-gal (right panel) in same sections. (**G**) Representative images show phospho-p38 positive (brown) cells and senescent (blue) cells. (**H**) Bar graph representing mean pixel intensity of p-p38 (left panel) and β-gal (right panel). (**I**) Representative images showing phospho-NFκB (brown) and senescent cells (blue) across the treatment group. (**J**) Bar graph showing average pixel intensity of p-NFκB (left panel) and SA-β-gal (right panel). Nuclei were counterstained with Hematoxylin. A FOV of at least 10 was used for statistical quantification and analysis. Scale bar, 10 μm. *, significant relative to control; **, significant relative to γ-ray.

**Figure 4 cancers-16-03392-f004:**
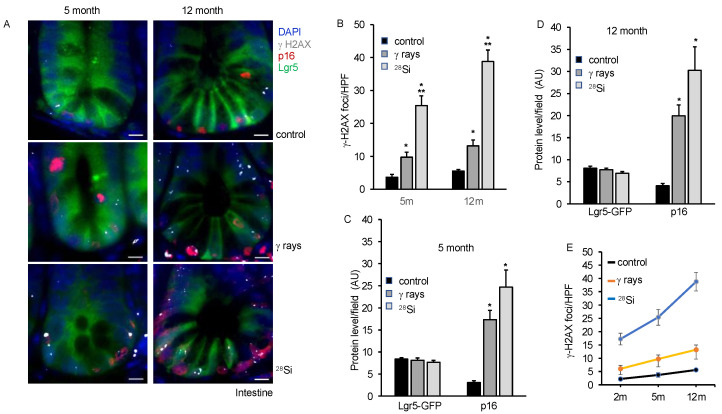
^28^Si exposure leads to persistent increased DNA damage and senescence at 5 or 12 months after irradiation. (**A**) Fluorescent images of γH2AX, p16, or Lgr5-GFP showing increased DNA damage and senescence in ISCs at 5 months (left panel) and 12 months (right panel) after radiation exposure. Nuclei were counterstained with DAPI. At least ten different FOVs were captured, γH2AX foci were counted and Lgr5-GFP fluorescent intensity was quantified. Scale bar, 10 μm. (**B**) γH2AX foci were counted and represented graphically, showing time-dependent increased DNA damage response. Graphical representation of p16 or Lgr5-GFP fluorescent intensity at 5 months (**C**) and 12 months (**D**) post radiation exposure. (**E**) Graphical representation of average γH2AX foci per HPF at 2 m, 5 m and 12 m post-irradiation in intestinal sections. *, significant relative to control; **, significant relative to γ-ray.

**Figure 5 cancers-16-03392-f005:**
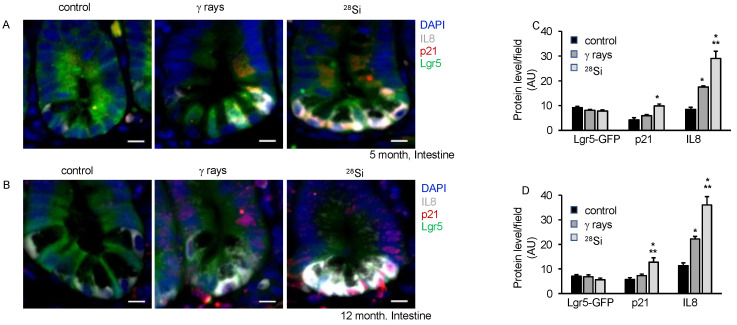
Space radiation exposure induces senescence and promotes SASP acquisition at the 5- and 12-month time points. Intestinal tissue sections were co-stained for IL8, p21, and Lgr5-GFP, and signals were detected under a fluorescent microscope. Representative fluorescent images of IL8, p21 and Lgr5-GFP staining showing senescence and SASP at 5 months (**A**) and 12 months (**B**) post-radiation. Nuclei were visualized using DAPI. Scale bar, 10 μm. At least ten different FOV images captured were fluorescent intensity quantified and represented graphically for 5 months (**C**) and 12 months (**D**) post-irradiation. *, significant relative to control; **, significant relative to γ-ray.

**Figure 6 cancers-16-03392-f006:**
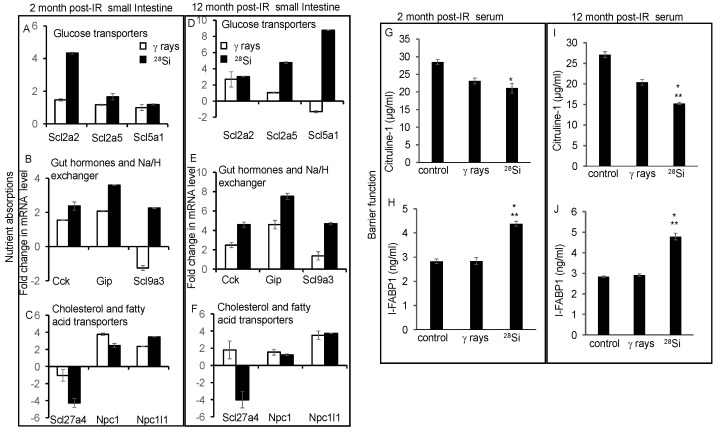
Space radiation exposure perturbs barrier functions and genes regulating nutrient absorption up to 1 year after irradiation. Bar graph showing relative mRNA levels of nutrient transporters at 2 months (**A**–**C**) and 12 months (**D**–**F**) after irradiation showing altered expression of glucose transporter, gut hormones and Na/H exchanger, cholesterol and fatty acid transporters genes. Serum citrulline levels were detected using an ELISA assay kit. Bar graph showing serum citrulline level (μg/mL) at 2 months (**G**) and 12 months (**I**) after irradiation. Serum IFABP was detected using an I-FABP1 ELISA assay kit. Bar graph showing levels of I-FABP1 (ng/mL) in serum at 2 months (**H**) and 12 months (**J**) after irradiation. *, significant relative to control; **, significant relative to γ-ray.

**Figure 7 cancers-16-03392-f007:**
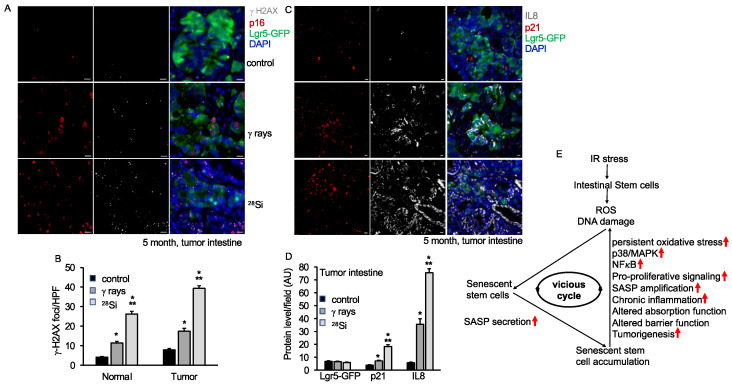
^28^Si exposure leads to persistent increased DNA damage, senescence and SASP after 5 months of radiation in intestinal tumor from *Lgr5*^+^*Apc*^1638N/+^ mice. Intestinal tumor sections were co-stained with p16, γH2AX, and Lgr5-GFP to detect DNA damage response and senescence after 5 months of radiation exposure. Representative fluorescent images show increased p16 and γH2AX expression in tumor tissue sections. (**A**) Representative fluorescent images showing increased senescence (p16) and DNA damage response (γH2AX) expression in tumor intestine tissue sections after 5 months of radiation exposure. (**B**) γH2AX foci were counted and represented graphically, showing higher DNA damage response in tumors relative to unirradiated control. Graphical representation of Lgr5-GFP or p16 fluorescent intensity in the tumor samples. (**C**) Representative fluorescent images of IL8, p21, and Lgr5-GFP in the tumor intestine show increased senescence and SASP activity after radiation. (**D**). The fluorescent intensity of IL8, p21, and Lgr5-GFP was quantified and represented as a bar graph. Nuclei were visualized using DAPI. (**E**) Schematic representation and summary of observations after space radiation exposure in mouse ISCs. Red arrow indicates increased levels after irradiation. Scale bar, 10 μm. *, significant relative to control; **, significant relative to γ-ray. Statistical significance is set at *p* < 0.05, and error bars represent mean ± SEM.

## Data Availability

All data is contained within the article.

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
