# Peer review of "Effects of High-Linear-Energy-Transfer Heavy Ion Radiation on Intestinal Stem Cells: Implications for Gut Health and Tumorigenesis"

_cancers, 2024, doi:10.3390/cancers16193392_

Round 1

Reviewer 1 Report

Comments and Suggestions for Authors

Thank you for the opportunity to review this manuscript. The manuscript “Effects of High-LET heavy ion radiation on intestinal stem cells: Implications for gut health and tumorigenesis” presents a very well conducted study. Seldom, I have reviewed manuscripts without many comments/suggestions. Here, I have only one suggestion about the discussion which is curiously short considering the detailed results presented in the paper. I would suggest discussing the presented details further. Also, a minor comment about replacing “Intestinal stem cells” in key words with suitable similar word. As this word has already been used in the title of the manuscript.

Author Response

Comments: Thank you for the opportunity to review this manuscript. The manuscript “Effects of High-LET heavy ion radiation on intestinal stem cells: Implications for gut health and tumorigenesis” presents a very well conducted study. Seldom, I have reviewed manuscripts without many comments/suggestions. Here, I have only one suggestion about the discussion which is curiously short considering the detailed results presented in the paper. I would suggest discussing the presented details further. Also, a minor comment about replacing “Intestinal stem cells” in key words with suitable similar word. As this word has already been used in the title of the manuscript.

Response: We appreciate your recognition of the significance of our work and the clarity of our writing. As suggested, we have elaborated the discussion and removed ‘Intestinal Stem Cells’ from the list of keywords.

Reviewer 2 Report

Comments and Suggestions for Authors

In this manuscript, the authors described gamma-ray and silicon beam irradiation experiments on mice to evaluate impact of heavy ionizing particle exposure on intestinal stem cells. The results were clear-cut and reasonable. Only minor requests for revision are:

  1. To explain how the radiation doses were determined. Adding reference papers would be helpful.
  2. To add reference, if any, showing long-lasting gamma-H2AX foci in other irradiation experiments. DNA damage response still observed 2 months after irradiation may be surprising for readers.

Author Response

Comments: In this manuscript, the authors described gamma-ray and silicon beam irradiation experiments on mice to evaluate impact of heavy ionizing particle exposure on intestinal stem cells. The results were clear-cut and reasonable. Only minor requests for revision are:

  1. To explain how the radiation doses were determined. Adding reference papers would be helpful.

Response: We thank the reviewer for reviewing our manuscript and providing us with valuable suggestions. The selection of a 50 cGy dose of was based on our previous studies [33-35]. Accurate dose delivery was ensured through beam calibration and dosimetry by NSRL physicists, with variability between exposures maintained within ±2.5%. For further details on beam calibration and dosimetry please refer to https://www.bnl.gov/nsrl/userguide/dosimetry-calibration.php.

  1. To add reference, if any, showing long-lasting gamma-H2AX foci in other irradiation experiments. DNA damage response still observed 2 months after irradiation may be surprising for readers.

Response: We thank the reviewer for the suggestion. We previously published a paper (Kumar et al, 2018; PNAS) demonstrating gamma-H2AX foci from 7 days up to 12 months post-exposure to gamma and particle radiation.  We have discussed this and added the reference in the revised manuscript as suggested.

Reviewer 3 Report

Comments and Suggestions for Authors

The work presented in this manuscript is quite thorough except there are some minor issues.

1.     Lines 61-63. The statement entitled “IR generates -----passes through [17].” is not a correct statement. In fact, the correct statement is “The best overall estimate of the probability of direct ionization at a given site in a molecule, e.g., in DNA - such as the sugar, phosphate, or DNA base, is provided by the number of valence electrons at that site (J. Phys. Chem. B 2012, 116, 5900−5906)”

2.     Lines 83-104. High HZE particles are known to cause clustered damage, release of short fragments etc. The authors should mention these here and cite these references: (a) RADIATION RESEARCH 145, 200-209 (1996), (b) Advances in Space Research 34 (2004) 1353–1357, (c) FREE RADICAL RESEARCH, 2016, VOL. 50, NO. S1, S64–S78. http://dx.doi.org/10.1080/10715762.2016.1232484, (d) Molecules 2022, 27, 1540. https://doi.org/10.3390/molecules27051540

3.     The experimental results shown in Figure 1 and discussed in sections 3.1 and 3.2 (the section no 3.2.28 is incorrect) could be qualitatively explained by the Monte Carlo calculations DNA 2024, 4, 34–51. https://doi.org/10.3390/dna4010002 (see Figure 6 in this paper). The authors should modify their discussions accordingly.

4.     The authors are requested to include the points 2 and 3 in the Conclusion of their paper.

Author Response

The work presented in this manuscript is quite thorough except there are some minor issues.

  1. Lines 61-63. The statement entitled “IR generates -----passes through [17].” is not a correct statement. In fact, the correct statement is “The best overall estimate of the probability of direct ionization at a given site in a molecule, e.g., in DNA - such as the sugar, phosphate, or DNA base, is provided by the number of valence electrons at that site (J. Phys. Chem. B 2012, 116, 5900−5906)” 

Response: We thank the reviewer for critically reviewing our manuscript and providing us with valuable suggestions.  We have carefully considered your suggestions and modified the statement as suggested for lines 61-63 in the revised manuscript and also added the suggested reference.

  1. Lines 83-104. High HZE particles are known to cause clustered damage, release of short fragments etc. The authors should mention these here and cite these references: (a) RADIATION RESEARCH 145, 200-209 (1996), (b) Advances in Space Research 34 (2004) 1353–1357, (c) FREE RADICAL RESEARCH, 2016, VOL. 50, NO. S1, S64–S78. http://dx.doi.org/10.1080/10715762.2016.1232484, (d) Molecules 2022, 27, 1540. https://doi.org/10.3390/molecules27051540

Response: We thank the reviewer for the insightful suggestion. We have modified the Lines 83-104 and incorporated the suggested lines.

  1. The experimental results shown in Figure 1 and discussed in sections 3.1 and 3.2 (the section no 3.2.28 is incorrect) could be qualitatively explained by the Monte Carlo calculations DNA 2024, 4, 34–51. https://doi.org/10.3390/dna4010002 (see Figure 6 in this paper). The authors should modify their discussions accordingly.

Response: We appreciate the reviewer for the valuable suggestion. We have corrected the section number and the discussion is modified accordingly in the revised text. We also included the citation of the suggested article.

  1. The authors are requested to include the points 2 and 3 in the Conclusion of their paper.

Response: We are very grateful for the positive feedback and thank the reviewer for the valuable suggestions which have helped us to improve the manuscript. We have included points 2 and 3 in the conclusion as suggested by the reviewer.